# Assessing multiple score functions in Rosetta for drug discovery

**Shannon T. Smith**[1,2], **Jens Meiler**[2,3,4]*

**1** Chemical and Physical Biology Program, Vanderbilt University, Nashville, Tennessee, United States of America, **2** Center for Structural Biology, Vanderbilt University, Nashville, Tennessee, United States of America, **3** Departments of Chemistry, Pharmacology, and Biomedical Informatics, Center for Structural Biology and Institute of Chemical Biology, Nashville, Tennessee, United States of America, **4** Institute for Drug Discovery, Leipzig University Medical School, Leipzig, Städelschule, Germany

* jens@meilerlab.org

## Abstract

Rosetta is a computational software suite containing algorithms for a wide variety of macro-molecular structure prediction and design tasks including small molecule protocols commonly used in drug discovery or enzyme design. Here, we benchmark RosettaLigand score functions and protocols in comparison to results of other software recently published in the Comparative Assessment of Score Functions (CASF-2016). The CASF-2016 benchmark covers a wide variety of tests including scoring and ranking multiple compounds against a target, ligand docking of a small molecule to a target, and virtual screening to extract binders from a compound library. Direct comparison to the score functions provided by CASF-2016 results shows that the original RosettaLigand score function ranks among the top software for scoring, ranking, docking and screening tests. Most notably, the RosettaLigand score function ranked 2/34 among other report score functions in CASF-2016. We additionally perform a ligand docking test with full sampling to mimic typical use cases. Despite improved performance of newer score functions in canonical protein structure prediction and design, we demonstrate here that more recent Rosetta score functions have reduced performance across all small molecule benchmarks. The tests described here have also been uploaded to the Rosetta scientific benchmarking server and will be run weekly to track performance as the code is continually being developed.

## Introduction

Computational structure-based drug discovery is a powerful strategy used to identify and optimize ligands. Computational protocols that study protein-small molecule (aka ligand) interactions are used in tandem with experiments in structure-based computer aided drug discovery (SB-CADD) [1–4]. A central aspect of SB-CADD methods is the development of an accurate score function to quantify the physicochemical interactions of protein-ligand complexes [5,6]. Score functions have been continually developed since their inception; however, reliably calculating interactions between a protein target and a small molecule remains a formidable challenge [7,8]. In terms of ligand docking of one small molecule to a target, the score function

**Data Availability Statement:** The Rosetta 3.10 software suite is publicly available and the license is free for non-commercial users at http://www.rosettacommons.org/. Method details can be found in the Rosetta Commons documentation and are included with Rosetta 3.10 under the directory

“demos/tutorials/ligand_docking” or downloadable from the Meiler Lab website. Any additional information reported here can be found in the attached Protocol Capture.

**Funding:** STS received funding for this work through the National Cancer Institute of the National Institutes of Health (F31 CA243353). STS also received funding through the PhRMA Foundation's Pre-Doctoral Fellowship in Informatics (phrmafoundation.org). The funders had no role in study design, data collection and analysis, decision to publish, or preparation of the manuscript.

**Competing interests:** The authors have declared that no competing interests exist.

ideally extracts the native binding mode in comparison to non-native decoys [9]. Arguably a more difficult scoring problem is to estimate and rank binding energies across a compound series [10] or screen larger libraries to extract likely binders to given target [11]. Score functions largely fall into physics-based [12], empirical [13], knowledge-based [14] or machine learning [15] approaches. These calculations are a less resource-intensive approach in comparison to higher levels of theory and quantum-level calculations, albeit with a trade-off in accuracy [16,17]. For this reason, score functions are commonly used during early stage drug discovery and their development remains an active area of research.

The recent report of Comparative Assessment of Scoring Functions 2016 (CASF-2016) provides a standard benchmark for score function evaluation. CASF-2016 [18–20] measures the performance of different score functions across various tests. These tests include scoring and ranking of multiple compounds against the same target to assess a score function's ability to favor tighter binders, docking of one compound to a respective protein target structure to assess how well a score function discriminates against incorrect binding modes, and screening for known binders against within a library of non-binding decoys (Fig 1). CASF-2016 organizers focus on the score function without introducing sampling biases, therefore they provide all structures in the dataset, as well as decoys for docking and screening tests. It is important to explicitly state that all runs and analyses are performed using from the distributed CASF-2016 inputs and scripts, allowing for direct comparisons to published results of other protocols. We exclude the $\Delta_{Vina}RF_{20}$ [21] from our analysis because ~50% of the dataset was included in the training set during development of this score function, as noted in [18]. Confidence intervals are also calculated using a provided bootstrapping method as described in CASF-2016.

While the terms of the Rosetta score functions are similar to terms used in molecular mechanics, the shape of the potential is mostly determined from statistics over geometries observed in experimentally determined structures deposited in the Protein Data Bank (PDB) (www.rcsb.org) [22]. It is assumed that structural features commonly observed in the PDB knowledge base are energetically more favorable. By using a Boltzmann equation, this

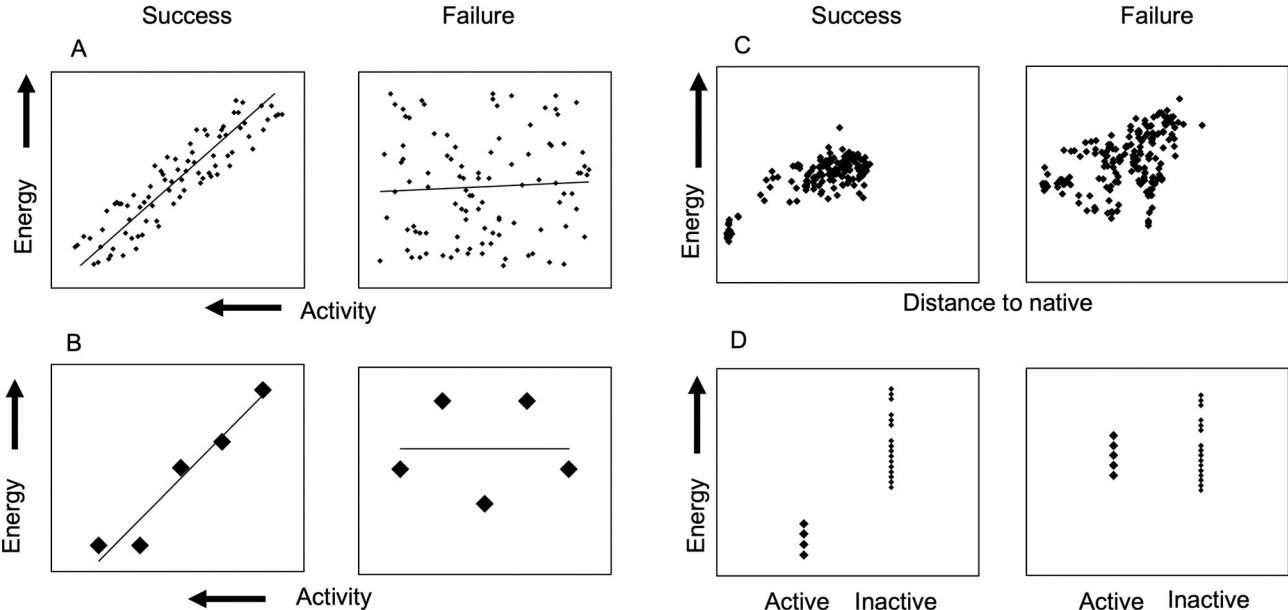

**Fig 1. Schematic of CASF-2016 tests.** Examples of success and failures in (A) Scoring test; (B) Ranking test; (C) Docking test and (D) Screening test.

probability is converted into a pseudo energy. Although these score terms are largely inspired by statistics, the corresponding weights have been optimized to some extent to align with experimentally observed free energies. Here, we provide a brief overview of score functions throughout Rosetta development; a comprehensive score function overview is presented in [23,24].

The score functions tested here are RosettaLigand [25,26], Talaris2014 [27–29], Ref2015 [23,24] and Betanov16 [30]. Briefly, the RosettaLigand score function is a slight reweight to the Score12 score function [31] whereas Talaris2014, Ref2015 and Betanov16 are consecutive extensions of Score12. Note that there have been underlying changes between score functions with regards to weight refitting, including energy well-depths and other atom-specific parameters, but this overview will only mention major score term changes.

The Score12 Rosetta score function is a linear summation of score terms consisting of Lennard-Jones Van der Waals potentials, Lazaridis-Karplus implicit solvation penalties, orientation-dependent hydrogen bonding, residue pair potentials, statistically-derived side chain rotamer and backbone torsional energies, and a reference energy for each amino acid representing the energy of the unfolded state (used in protein design), each with respective weights [31]. The RosettaLigand score function is similar to Score12, with the addition of score terms for small molecule ligands and a Coulombic pair-wise electrostatic term [25]. The Talaris2014 score function was developed from Score12 with the replacement of the residue pair potential with a Coulombic term to more accurately represent both short- and long-range electrostatic representations and implementation of the Dunbrack 2010 rotamer library [27–29]. Derived from the Talaris2014 score function, Ref2015 includes desolvation penalty terms with orientation-dependent solvation of polar atoms [23,24]. Betanov16 is largely similar to Ref2015 with parameter re-training to improve polar interaction characterization and an implicit water bridging term [30].

This paper reports the performance of multiple score functions used over the course of Rosetta development in CASF-2016 tests. The purpose of this assessment is three-fold. First, we want to fully benchmark different Rosetta scoring functions for these SB-CADD tasks as they developed over time. We will determine the current 'best-practice' in using Rosetta for SB-CADD. For this purpose, this benchmark tests the RosettaLigand [25], Talaris2014 [27], Ref2015 [23,24] and Betanov16 [30] score functions. Second, we want to establish benchmarks that allow us to track changes in the code that either enhance or worsen performance and use this as a directive for future protocol and score function development. Third, a direct comparison of Rosetta score functions to other software reported in CASF-2016 suggests advantages and disadvantages to using Rosetta for specific research questions. Recent Rosetta score function development has focused heavily on optimizing for canonical protein structure prediction and design protocols; however, there has been little development in the score function with regards to small-molecule applications.

In addition to CASF-2016 tests, we also perform ligand docking with full sampling to investigate performance differences between Rosetta score functions in a more realistic test-case for users. Ligand docking with full sampling uses the standard Rosetta protocol which allows extensive flexible binding pocket sampling to obtain diverse ligand poses while using a score function to correctly identify the native binding mode. The purpose of this test is to determine how well each score function favorable scores the native binding mode against non-native decoy poses. Importantly, using poses sampled by Rosetta instead of pre-selected binding modes by CASF-2016 allows us to extract specific protein-ligand interactions considered favorable or unfavorable in each score function. Using these insights along with results from the other diverse CASF-2016 tests, this provides a more thorough benchmark to guide future work.

## Results

### Starting structures for each benchmark are provided by CASF-2016 datasets

CASF-2016 provides all structural input data for each benchmark test in aims to reduce sampling biases between score functions and allowing for direct comparison between methods. In the scoring and ranking tests, which use a known structure to correlate score with experimental binding, CASF-2016 provides "raw" structures directly from the Protein Data Bank [22] without any optimization or structure modification. All Rosetta score functions use the Leonard-Jones 6–12 repulsive potential, which is sensitive to slight clashing and simply scoring unrefined structures results in artificially high values, clearly artifacts of this repulsive score term. As mentioned in CASF-2016 paper, score functions sensitive to clashing require optimization to alleviate artifacts therefore we minimize the structures prior to scoring. CASF-2016 also provides a locally-optimized "opt" set meant to slightly tweak the structure by minimizing significant clashes or unrealistic geometries. It should be noted that some inputs from the given opt set still have severe clashing therefore we perform the same minimization protocols on both the raw and opt inputs. Comparison of these results showed little change between raw and opt sets across various protocols and score functions (S1 Fig). Analysis presented here uses the raw set inputs, as this is more applicable to typical use-cases.

For each benchmark test, we present protein subset results according to the CASF-2016 defined criteria to understand the effects of the binding pocket environment on performance, specifically the size, solvent exposure and hydrophobicity of surrounding residues. Subsets are parsed by three metrics: pocket hydrophobicity (H), solvent-accessible surface area of the ligand molecule buried upon binding (ΔSAS), and excluded volume inside the binding pocket upon ligand binding (ΔVOL). CASF-2016 parsed structures into three bins for each metric, each containing 19 protein-ligand clusters, denoted as H1/H2/H3, S1/S2/S3, and V1/V2/V3. H1 contains more polar pocket residues, S1 contains more solvent-exposed ligand pockets, and V1 contains smaller binding pockets.

### Rosetta comparison to other CASF-2016 reported score functions

Across the CASF-2016 tests, Rosetta compared well to other reported score functions from other software (Fig 2). Most notably, the RosettaLigand score function is 2/34 when testing ability to rank multiple compounds against the same target. Most score functions perform similarly and not well in the scoring test with correlation ranging from 0.4–0.6, demonstrating the challenge of comparing across larger chemical space and normalizing diverse protein-ligand interactions. The ranking test shows reasonable ability to rank multiple compounds to a single target. In the docking test, score functions can distinguish a near-native binding pose when given a non-native decoys. All score functions tested here and in CASF-2016 perform poorly when tested to extract known binders against a library of non-binders, demonstrating the difficulty of reliable virtual screening across large compound sets.

### Newer Rosetta score functions show decreased performance in all CASF-2016 tests

In each test, we demonstrate that the newer score functions, which have been heavily optimized for protein structure prediction and design, perform worse in comparison to the original RosettaLigand score function for different small molecule tasks (Fig 3). The differences are

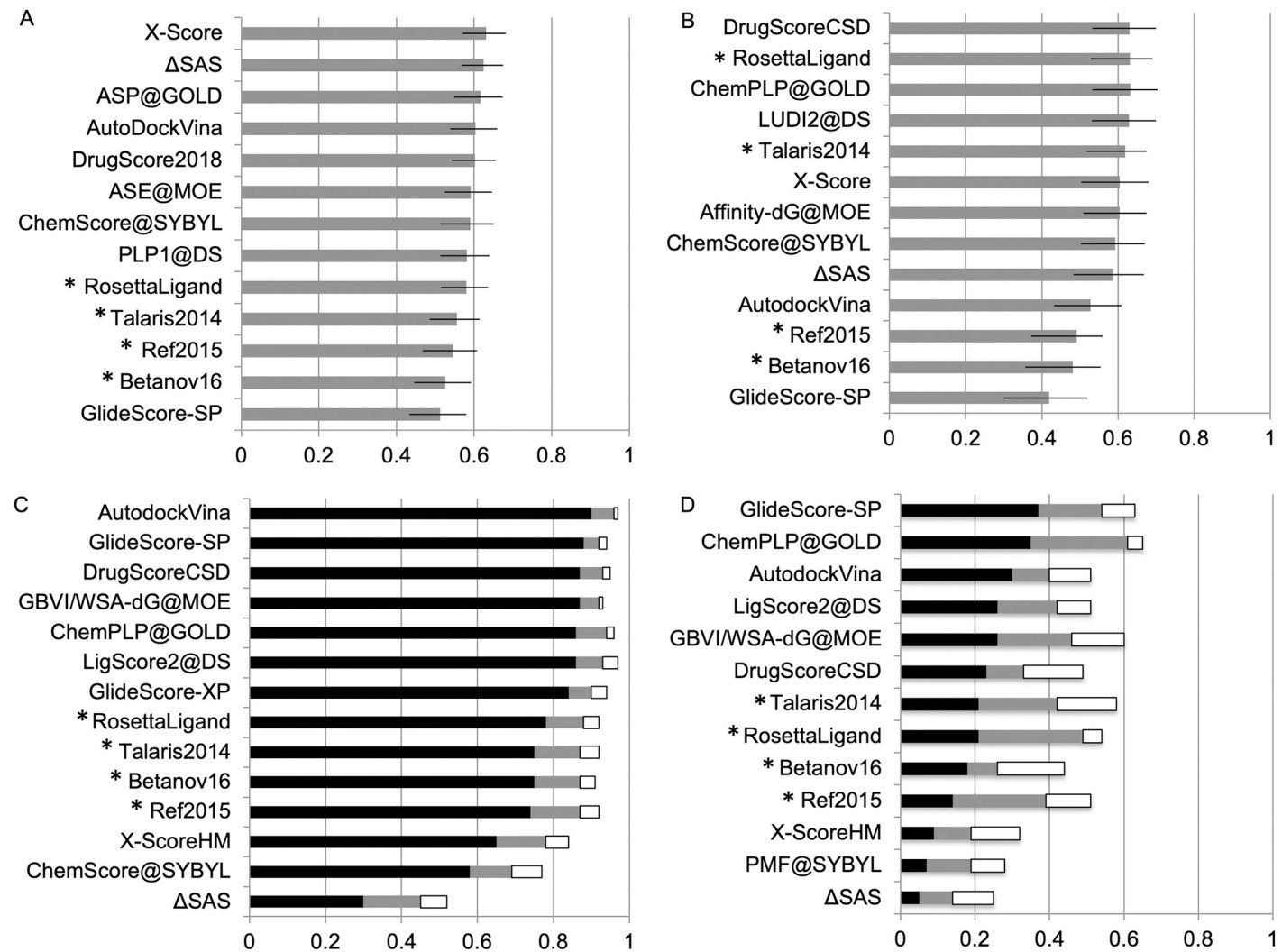

**Fig 2. Results of Rosetta score functions and other reported software in CASF-2016.** (A) Pearson correlation in scoring test. (B) average Spearman correlation in ranking test. (C) Docking test with percentage of complexes where native pose within top 1%, 2% and 3% shown in black, gray and white, respectively. (D) Forward-screening test with percentage of complexes scoring known binder in top 1%, 5% and 10% shown in black, gray and white, respectively. Confidence intervals shown in (A) and (B). For simplicity, only the top performing score function from each external program is presented here.

small in the scoring, docking and screening tests; however, the difference is most notable in the ranking test where RosettaLigand and Talaris2014 average correlations hover at 0.6, whereas Ref2015 and Betanov16 at 0.5. It should also noted that computation speeds are nearly doubled when using the newer score functions, an issue that is currently being addressed in the developer community. Results of each Rosetta score function are discussed below and summarized in Table 1.

**Scoring test results.** The scoring test uses the Pearson correlation to analyze a score function's ability to linearly correlate the predicted interaction with experimental binding affinity across the full 285 protein-ligand complex dataset. Pearson correlation varies between 0 and 1 (where 1 is a perfect correlation) and is calculated by (1) where $x_i$ and $y_i$ are the binding score

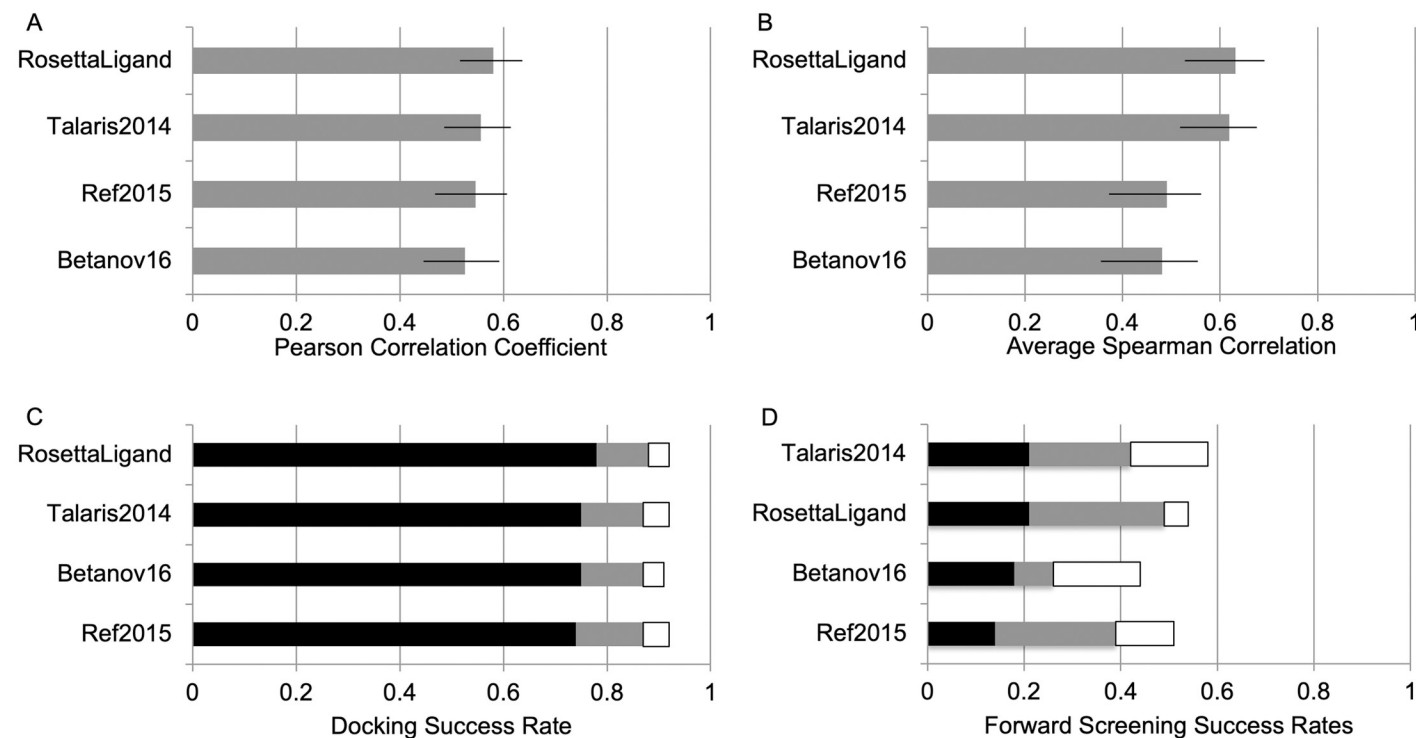

**Fig 3. Results of the different Rosetta score functions.** (A) Pearson correlation in scoring test. (B) average Spearman correlation in ranking test. (C) Docking test with percentage of complexes where native pose within top 1%, 2% and 3% shown in black, gray and white, respectively. (D) Forward-screening test with percentage of complexes scoring known binder in top 1%, 5% and 10% shown in black, gray and white, respectively. Confidence intervals shown in (A) and (B).

**Table 1. Subset results across all CASF-2016 tests.** All: Across full dataset, H1-3 hydrophobicity scale with H3 being most hydrophobic pocket; S1-3 with S3 being most solvent-exposed; V1-3 with V3 being largest pocket volume.

| | | All | H1 | H2 | H3 | S1 | S2 | S3 | V1 | V2 | V3 |
|---|---|---|---|---|---|---|---|---|---|---|---|
| Scoring | RosettaLigand | 0.58 | 0.56 | 0.66 | 0.50 | 0.66 | 0.45 | 0.59 | 0.32 | 0.50 | 0.64 |
| | Talaris2014 | 0.56 | 0.53 | 0.62 | 0.48 | 0.61 | 0.40 | 0.64 | 0.22 | 0.52 | 0.65 |
| | Ref2015 | 0.55 | 0.49 | 0.58 | 0.56 | 0.60 | 0.43 | 0.59 | 0.23 | 0.50 | 0.60 |
| | Betanov16 | 0.53 | 0.50 | 0.58 | 0.49 | 0.61 | 0.41 | 0.56 | 0.23 | 0.46 | 0.58 |
| Ranking | RosettaLigand | 0.63 | 0.58 | 0.62 | 0.70 | 0.74 | 0.63 | 0.52 | 0.73 | 0.49 | 0.68 |
| | Talaris2014 | 0.62 | 0.51 | 0.63 | 0.72 | 0.67 | 0.65 | 0.52 | 0.72 | 0.47 | 0.67 |
| | Ref2015 | 0.49 | 0.51 | 0.49 | 0.48 | 0.56 | 0.60 | 0.28 | 0.54 | 0.41 | 0.53 |
| | Betanov16 | 0.48 | 0.53 | 0.49 | 0.42 | 0.54 | 0.59 | 0.28 | 0.52 | 0.41 | 0.52 |
| Docking | RosettaLigand | 0.78 | 0.78 | 0.81 | 0.76 | 0.70 | 0.80 | 0.86 | 0.67 | 0.84 | 0.83 |
| | Talaris2014 | 0.75 | 0.79 | 0.79 | 0.68 | 0.71 | 0.76 | 0.80 | 0.66 | 0.82 | 0.78 |
| | Ref2015 | 0.74 | 0.78 | 0.68 | 0.76 | 0.67 | 0.76 | 0.79 | 0.68 | 0.78 | 0.76 |
| | Betanov16 | 0.75 | 0.82 | 0.70 | 0.73 | 0.68 | 0.75 | 0.81 | 0.66 | 0.83 | 0.75 |
| Screening | RosettaLigand | 0.21 | 0.16 | 0.32 | 0.16 | 0.21 | 0.19 | 0.24 | 0.11 | 0.26 | 0.26 |
| | Talaris2014 | 0.21 | 0.26 | 0.26 | 0.11 | 0.16 | 0.24 | 0.24 | 0.00 | 0.32 | 0.32 |
| | Ref2015 | 0.14 | 0.16 | 0.21 | 0.05 | 0.16 | 0.14 | 0.12 | 0.05 | 0.16 | 0.21 |
| | Betanov16 | 0.18 | 0.21 | 0.26 | 0.05 | 0.16 | 0.19 | 0.18 | 0.05 | 0.21 | 0.26 |

and experimental binding constant of a complex, respectively.

$$R_{Pearson} = \frac{\sum_i^n (x_i - \bar{x})(y_i - \bar{y})}{\sqrt{\sum_i^n (x_i - \bar{x})^2}\sqrt{\sum_i^n (y_i - \bar{y})^2}} \tag{1}$$

In the scoring test, Rosetta score functions perform similarly to others with Pearson correlations hovering between 0.5 and 0.6. There are only slight changes in performance depending on the score function used, which prompt more specific analyses on subsets based on protein and ligand properties. We observe improved performance in binding pockets with moderate hydrophobicity (subset H2) compared to those that are more polar (H1) or hydrophobic (H3). Solvent accessibility decreases performance in pockets with moderate exposure to solvent (S2) compared to surface pockets with large solvent exposure (S1) and deeper pockets with little solvent exposure (S3). Correlation in complexes containing small pocket volumes (V1) drops significantly compared to larger pocket volumes (V2, V3). This is a frequent observation across many tested score functions in the CASF-2016 analysis; however, this effect is more apparent with Rosetta where V1 correlations ranged from 0.2–0.3 compared to ~0.5 and ~0.6 for V2 and V3 subsets, respectively. The RosettaLigand score function performs similarly to other Rosetta score functions for V2 and V3 subsets; however, it is markedly better in the V1 subset.

**Ranking test results.** The ranking test uses the Spearman correlation to analyze a score function's ability to rank binding affinities of compounds against the same protein target. The starting 285 complexes are partitioned into 57 clusters each containing 5 protein-ligand pairs with target sequence similarity > 90%, as well as the respective binding affinities of each ligand to its respective target. The Spearman correlation of each system varies between 0 and 1 (where 1 is perfect ranking) and is calculated as in (2) where $rx_i$ and $ry_i$ are the ranks of the binding score and experimental binding constant of the $i$th complex, respectively, and $n$ is the number of samples in the complex.

$$\rho_{Spearman} = \frac{\sum_i^n (rx_i - \overline{rx})(y_i - \overline{ry})}{\sqrt{\sum_i^n (x_i - \overline{rx})^2}\sqrt{\sum_i^n (y_i - \overline{ry})^2}} \tag{2}$$

In the ranking test, we obtain an average Spearman correlation of 0.63 using the RosettaLigand score function, which performs similarly to DrugScoreCSD, ChemPLP@GOLD and LUDI2@DS. An important observation from this study is the difference in ranking ability of the Rosetta score functions, specifically between the RosettaLigand and Talaris2014 (0.60–0.63) score functions compared to Ref2015 and Betanov16 (0.48–0.51). For RosettaLigand and Talaris2014 score functions, we see steady increasing performance in increasing hydrophobic environments (H3>H2>H1). Performance is similar in Ref2015 across hydrophobic environments and performance decreases in increasing hydrophobicity in Betanov16. Solvent exposure analysis reveals performance decreases for all score functions in more solvent occluded pockets (subset S3). The results are particularly drastic for Ref2015 and Betanov16 score functions in less solvent exposed pockets where we see ~0.6 and ~0.3 for the S2 and S3 subsets, respectively. As mentioned, there is drop-off in performance in less solvent accessible pockets for the RosettaLigand and Talaris2014 score functions as well; however, correlations are 0.60 and 0.53 for S2 and S3 subsets, respectively. Across all score functions, pockets of moderate pocket volume size (V2) perform worse than complexes with small/large pocket volumes (V1, V2) and the large pocket volume subset perform best. Although this trend is observed across all score functions, RosettaLigand and Talaris2014 consistently have the highest correlations in each volume subset.

**Docking test results.** The docking test analyzes a score function's ability to distinguish true ligand binding orientation from decoys and is performed across the full 285 protein-ligand dataset. The provided docking test dataset contains 100 near native poses and decoys spanning from 0–10 Å RMSD relative to the co-crystallized structure for all 285 protein-ligand complexes. RosettaLigand and Talaris2014 score functions display an improved performance in distinguishing near-native poses, but these values range only between 75–78% of complexes across all tested score functions. Subset analysis shows moderate increases in performance in more polar pockets across all score functions and is particularly apparent in Ref2015 and Beta-nov16 score functions. All score functions perform better in increasingly non-solvent exposed pockets with success rates averaging at 69%, 77% and 82% for S1, S2 and S2 subsets, respectively. Docking into small pockets performs worse than medium-large pockets where we observe success rates of 67%, 82% and 78% in V1, V2 and V3 subsets, respectively.

**Screening test results.** The screening test analyzes a score function's ability to extract true binders for a protein target against a library of non-binders and is performed using one representative protein target from each of the 57 clusters derived in the ranking test against all 285 ligands from the full dataset. All 285 ligands are docked 100 times to each cluster representative. The 5 ligands belonging to that cluster are classified as true binders, whereas compounds from other clusters are considered non-binders. There are several known binding crossovers between the given clusters, which are taken into account in the analysis. RosettaLigand and Talaris2014 score functions successfully identify the highest-affinity binder within the top 1% of scored outputs in 21% of cases and an enrichment factor between six and seven for outputs within the top 1%. In all score functions tested here, enrichment values decreases as more compounds outside the top 1% were used, indicating that there is some ability to discriminate binders and non-binders. These effects are modest at best with maximum enrichments of 6.9, 4.4 and 3.1 using the top 1%, 5% and 10% of scored compounds, respectively. There are some general trends across all the score functions tested; however, there is considerable noise in these screening tests. We see overall performance drop-offs in largely hydrophobic pockets (H3) whereas the effects are less apparent in more polar pockets (H1, H2). Performance appears largely independent of solvent accessibility where we see little change across S1, S2 and S3 subsets. There is a considerable dependence on pocket volumes, notably the low success rates in complexes with small pocket volumes (V1).

## Full sampling docking protocols in Rosetta assess score functions in a test resembling typical use-cases

The full docking protocol sequentially performs coarse-grained low-resolution sampling, a high-resolution sampling step followed by a final energy minimization [32,33]. This protocol also performs extensive ligand conformer sampling; however, conformers must be pre-generated using another software such as BCL::Conf [34]. For each complex, we generated 1000 output poses, each with a corresponding score and RMSD relative to the experimentally determined co-crystal structure. Although we can't directly compare full docking results to those in CASF-2016, this allowed us to compare different internal Rosetta score functions on this dataset in a more realistic docking situation. This analysis uses the "$p_{near}$" metric defined by (3) to quantify a score function's ability to recapture the experimentally determined binding pose or the "funnel-likeness" of a RMSD vs. energy plot [35]. The $p_{near}$ calculation (3) is typically used to determine folding propensity where a value of 1 suggests a protein exists solely in the native state (0 Å), whereas a $p_{near}$ of 0 suggests a protein will never exist in the native state. Fig 1C example shows $p_{near}$ values of 1 and 0.1 for the left and right score vs. RMSD plots,

respectively.

$$p_{near} = \frac{\sum_{i=1}^{n} e^{-(RMSD_i^2/\lambda^2)} e^{-(E_i/kBT)}}{\sum_{j=1}^{n} e^{-(E_i/kBT)}}$$

(3)

where λ = 1.5 and kBT = 0.62.

## RosettaLigand outperforms other score functions in ligand docking with full sampling test

We run low-resolution sampling first to acquire the same starting positions before running high-resolution docking and final minimization in each score function. Comparison of number of near-native poses is the same across all score functions suggesting changes are due to scoring differences and not sampling bias (S2 Fig). We determine success of a docking run using the $p_{near}$ metric as described above and compare this value for each protein-ligand complex across all score functions. Comparison of $p_{near}$ values shows that overall, RosettaLigand scoring has greater $p_{near}$ values compared to Talaris2014, Ref2015 and Betanov16 (Fig 4). Points below the diagonal represent complexes where RosettaLigand outperformed the other score function, points above the diagonal represent RosettaLigand performed worse and points along the diagonal represent similar performance of both score functions. The number of cases where a $p_{near}$ difference is greater than 0.3 are shown.

## We test across a range of preparation and run protocols to document the best practices for specific tasks

Using these "raw" input structures, we run these through different preparation protocols to determine the best performing methods. Although there are many approaches to take, we narrowed down to how much protein refinement and ligand preparation is necessary prior to calculations for optimal performance. To test different protein preparation methods, we compare results using the raw structure directly downloaded from the PDB and a constrained-relax protocol on the apo protein structure [36]. We also report the effect of two partial charge generation methods, a python script within the Rosetta package (*molfiletoparams.py*) that assigns partial charges based on atom-type, and AM1-BCC partial charge methods using AMBER [37]. As mentioned previously, there are structures where small molecules with significant

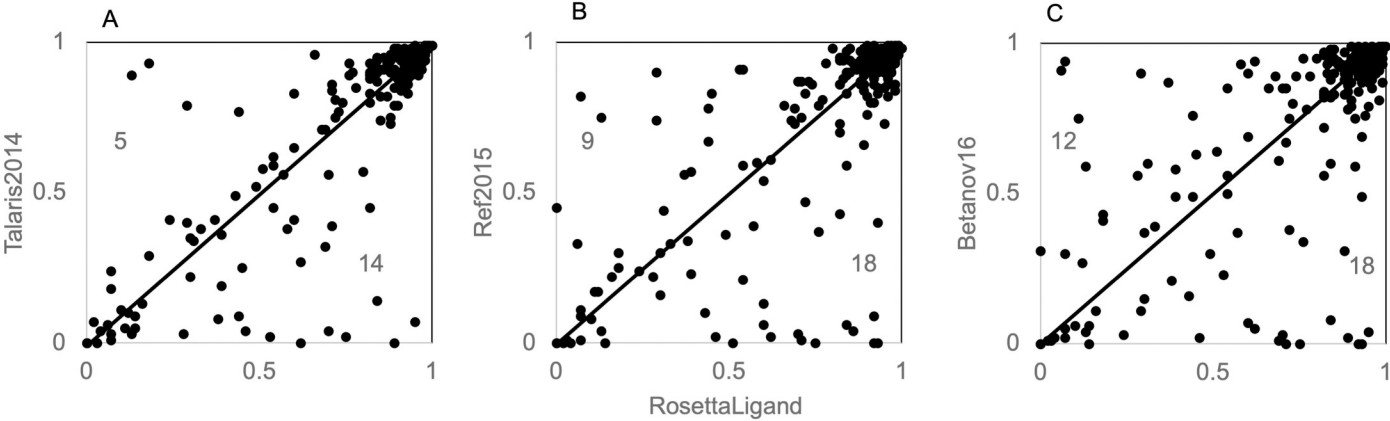

**Fig 4. $P_{near}$ values from the ligand docking with full sampling comparing RosettaLigand to other score functions.** Comparison of RosettaLigand performance to (A) Talaris2014, (B) Ref2015 and (C) Betanov16.

steric clashing artificially inflate Rosetta scores. We test whether simple minimization using the FinalMinimizer mover is enough to alleviate these clashes or if undergoing high-resolution sampling using the HighResDocker mover followed by the FinalMinimizer is better to estimate binding energies where an experimentally determined structure is provided. Scores were calculated using the InterfaceScoreCalculator mover. Changes were done in the RosettaScripts interface [38]. Each preparation pipeline was tested for all score functions. Based on these results, there is no significant change in ranking ability based on preparation schemes prior to scoring the complex (S3 Fig). The changes between the different score functions are consistent with prior findings where RosettaLigand/Talaris2014 outperform Ref2015/Betanov16 in all preparation settings.

## Discussion

### Testing across a range of inputs, preparation protocols and score functions leads to areas of focus in future development

The lack of change between ranking test results under different protocol pipelines signifies potential areas where Rosetta score functions need improvement. The lack of performance change between partial charging schemes, molfiletoparams and AM1-BCC, signifies that electrostatic terms need potential re-evaluating for small-molecule tasks. Another area to investigate is the solvation term, as this is one of the major changes between Talaris2014 and Ref2015 implementations. Pocket hydrophobicity analysis reveals important differences in ranking ability of each score function, showing that earlier score functions perform better in increasing hydrophobic environments, whereas newer score functions perform worse. These data suggest the need to potentially re-evaluate hydrophobic calculations for small molecules in the newer framework.

### We propose the immediate next steps for Rosetta development in SB-CADD

Rosetta contains functionality used in other protocols that can be translated to small-molecule predictions to aid binding predictions. A possible avenue is to apply protein-protein docking and interface analyses shown to improve prediction accuracy in other benchmarks to small molecule predictions [39,40]. The InterfaceAnalyzer Mover in RosettaScripts, initially written to analyze protein-protein interfaces, provides helpful tools to compare multiple binding poses [40,41]. Work towards a more ensemble-like representation of protein and a small molecule may also provide a more realistic calculation of a binding event. Unlike canonical amino acids, Rosetta currently does not contain chemical or physical information specific to small molecules and relies on external software to generate ligand parameters. Consequently, Rosetta doesn't properly score ligand energetics such as torsional strain. Next efforts also include the existing Rosetta framework to obtain ensemble representations of protein-ligand interactions for a representation more closely resembling natural binding events.

### This report brings to light the importance of constant, meaningful benchmarking as code is constantly being developed

Here we present the most extensive small-molecule benchmarking of different Rosetta methods and score functions to date and allow simple comparison to other software programs. Moreover, we conclude that recent Rosetta score functions have reduced performance across all benchmarks tested here. This is particularly evident in the ranking test, where the RosettaLigand and Talaris2014 achieve significantly correlations than Ref2015 and Betanov16. The

results presented here are somewhat expected, as most work in recent Rosetta development has focused on improving protein structure prediction and design. Importantly, others have shown that newer Rosetta score functions have improved canonical protein structure prediction and design within the Rosetta framework. Importantly, this analysis is meant to capture the status of small molecule-related protocols within Rosetta while not discrediting these other findings. This paper should also begin the conversation within the developer sphere about using different score functions depending on the specific application. This is not a novel concept, as most other software suites already do this with score functions and force field optimization. The benchmarks described here have been added to the Rosetta scientific benchmarking server and will be run weekly to track performance as the code is continually being developed.

## Conclusion

This paper investigates the use of multiple score functions within the Rosetta framework to determine the best available methods for small molecule protocols. To accomplish this, we use the CASF-2016 benchmark set consisting of a range of tasks and an additional a docking run with full sampling. Direct comparison to the 34 score functions provided by CASF-2016 results shows that the original RosettaLigand score function performs well in comparison to other top software. Notably RosettaLigand is 2$^{nd}$ among other CASF-2016 participants when ranking multiple compounds against a single protein target, demonstrating potential use in compound optimization schemes. We also show that more recent Rosetta score functions that have focused on canonical protein structure and design have reduced performance across all protocols tested here. This dataset and scoring, ranking and docking tests have been uploaded to the Rosetta scientific benchmarking server for weekly testing to track performance as the code is continually being developed. By testing and developing our score functions against various parameters, we can gain perspective as to where our algorithms fail and require immediate attention moving forward.

## Supporting information

**S1 Fig. Score comparison between "raw" and "opt" sets for each protocol and score function.** Scores from raw and opt set are depicted on the x and y axes, respectively.
(TIF)

**S2 Fig. Percentage of sub-2 Å RMSD structures in docking test.** RosettaLigand value on the x-axis and Talaris2014 (A), Ref2015 (B) and Betanov16 (C) on the y-axis.
(TIF)

**S3 Fig. Ranking test results across Rosetta protocols.** Average Spearman correlation across multiple preparation protocols: un-optimized PDB file "raw" and constrained relax "relaxed" protein preparations, molfiletoparams.py script "molfiletoparams" and AM1-BCC "am1bcc" partial charge generation, and high-resolution sampling and final minimization "hires_finalmin" vs. only final minimization "finalmin". Each combination was tested within each score function.
(TIF)

**S1 File. Protocol capture.**
(DOCX)

## Acknowledgments

We thank Rocco Moretti for helpful discussion and technical help and all members of the Rosetta Commons community.

## Author Contributions

**Conceptualization:** Shannon T. Smith, Jens Meiler.

**Data curation:** Shannon T. Smith.

**Formal analysis:** Shannon T. Smith, Jens Meiler.

**Funding acquisition:** Shannon T. Smith.

**Investigation:** Shannon T. Smith.

**Methodology:** Shannon T. Smith, Jens Meiler.

**Supervision:** Jens Meiler.

**Validation:** Shannon T. Smith.

**Writing – original draft:** Shannon T. Smith.

**Writing – review & editing:** Shannon T. Smith, Jens Meiler.

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
