## [Decision Letter · Decision Letter 0]

18 Aug 2020

PONE-D-20-17584

Assessing multiple score functions in Rosetta for Drug Discovery

PLOS ONE

Dear Dr. Smith,

Thank you for submitting your manuscript to PLOS ONE. After careful consideration, we feel that it has merit but does not fully meet PLOS ONE’s publication criteria as it currently stands. Therefore, we invite you to submit a revised version of the manuscript that addresses the points raised during the review process.

The reviewers identified a few minor issues with the manuscript, which they found otherwise suitable for publication in our journal.

We look forward to receiving your revised manuscript.

Kind regards,

Emilio Gallicchio, Ph.D.

Academic Editor

PLOS ONE

Journal Requirements:

Reviewers' comments:

Reviewer's Responses to Questions

**Comments to the Author**

1. Is the manuscript technically sound, and do the data support the conclusions?

Reviewer #1: Yes

Reviewer #2: Yes

2. Has the statistical analysis been performed appropriately and rigorously? 

Reviewer #1: Yes

Reviewer #2: Yes

3. Have the authors made all data underlying the findings in their manuscript fully available?

Reviewer #1: Yes

Reviewer #2: Yes

4. Is the manuscript presented in an intelligible fashion and written in standard English?

Reviewer #1: Yes

Reviewer #2: Yes

5. Review Comments to the Author

Reviewer #1: The study outlines the assessment of several Rosetta energy functions for the RosettaLigand method. This study was necessary since the recent versions of the score functions show reduced performance across small molecule benchmarks and an appropriate dataset from CASF-2016 was selected to test this. Overall, the manuscript covers this topic in depth and with the expected detail in examination, my recommendation is to accept with revisions.

There are three points that I would like the authors to address :

1. The authors mention activity and binding as interchangeable terms. Rosetta or any other platform is unlikely to measure activity per se and energies measured are a more direct measure of binding. Appropriate correction/ discussion of this topic should be included to avoid misuse of this method.

2. Fig 2 - this figure does not specifically demonstrate how rosetta ranks 2/34 out of tested methods. Either a better graphical representation needs to be chosen or a better wording to explain this. The top 4 score functions seem to show similar values and error especially in graph B. Authors would need to further elaborate this point better.

3. In general, it would be good to discuss the biophysical significance of each score function chosen perhaps in describing how they score the same ligand bound target.

4. Typographical error: Line 59 these calculations are “a” resource intensive approach

Reviewer #2: Please see attached comments.

6. PLOS authors have the option to publish the peer review history of their article (what does this mean?). If published, this will include your full peer review and any attached files.

Reviewer #1: No

Reviewer #2: No

---

## [Author Response · Author response to Decision Letter 0]

23 Sep 2020

The authors are appreciative for the guidance from Reviewer's to improve this manuscript. The attached Response to Reviewers provides a detailed account for each individual suggestion that was given.

---

## [Editor Report · Decision Letter 1]

28 Sep 2020

Assessing multiple score functions in Rosetta for Drug Discovery

PONE-D-20-17584R1

Dear Dr. Smith,

We’re pleased to inform you that your manuscript has been judged scientifically suitable for publication and will be formally accepted for publication once it meets all outstanding technical requirements.

Kind regards,

Emilio Gallicchio, Ph.D.

Academic Editor

PLOS ONE

---

## [Editor Report · Acceptance letter]

30 Sep 2020

PONE-D-20-17584R1 

Assessing multiple score functions in Rosetta for drug discovery 

Dear Dr. Smith:

I'm pleased to inform you that your manuscript has been deemed suitable for publication in PLOS ONE. Congratulations! Your manuscript is now with our production department. 

Kind regards, 

on behalf of

Dr Emilio Gallicchio 

Academic Editor

PLOS ONE